# Infection of endothelial cells by *Streptococcus agalactiae* reveals potential role of PI-2b pilus on endothelial barrier dysfunction

Jessica Silva Santos de Oliveira¹, Bruna Alves da Silva Pimentel¹, Leonardo Nagao Ferreira², Maria Eduarda Negreiro e Silva², Gabriela da Silva Santos¹, Prescilla Emy Nagao¹/⁺

¹Universidade do Estado do Rio de Janeiro, Instituto de Biologia Roberto Alcantara Gomes, Laboratório de Biologia Molecular e Fisiologia de Estreptococos, Rio de Janeiro, RJ, Brasil
²Fundação Técnico-Educacional Souza Marques, Escola de Medicina Souza Marques, Rio de Janeiro, RJ, Brasil

**BACKGROUND** *Streptococcus agalactiae* is responsible for sepsis and meningitis, and the major cause of neonatal morbidity and mortality. However, how *S. agalactiae* disrupts endothelial barriers is poorly understood.

**OBJECTIVES** Analyse the influence of endothelial cell (HUVECs) growth under static and shear stress conditions during infection with *S. agalactiae*, and the role of pilus PI-2b during endothelial barrier disruption and increased endothelial permeability.

**METHODS** HUVECs under static and shear conditions were infected by *S. agalactiae* (GBS90356 and GBS90356Δ*pilus2b*) strains in the presence and absence of fibrinogen. VE-cadherin was evaluated by immunofluorescence and RT-PCR assays, and the endothelial permeability by transwell assay.

**FINDS** Shear stress induced the alignment of HUVECs and increased the adherence of *S. agalactiae* strains (GBS90356 and GBS90356Δ*pilus2b*), mainly in the presence of fibrinogen, in addition to greater peripheral localisation of VE-cadherin. Rupture points and damage to endothelial integrity was visualised after infection with the GBS90356WT strain, mainly in the presence of fibrinogen. RT-PCR analyses identified increase in VE-cadherin expression in HUVECs under shear stress and a decrease in VE-cadherin after infection, with increased levels of endothelial permeability.

**MAIN CONCLUSION** Data demonstrate for the first time the dysfunction of the adhesive barrier induced by the *S. agalactiae* ST-17 strain, mainly in HUVECs under shear stress, where PI-2b expression was essential to optimise the damage to endothelial integrity.

Key words: *Streptococcus agalactiae* - endothelial cells - fibrinogen - VE-cadherin - cell permeability

*Streptococcus agalactiae* is an important human pathogen with impacts on health and economy worldwide, being the main cause of maternal and neonatal infections, including chorioamnionitis, postpartum endometritis, pneumonia, sepsis and meningitis. *S. agalactiae* causes at least 400,000 maternal and neonatal infections, accounting for approximately 50,000 stillbirths and 50,000 to 100,000 infant deaths annually.[1] In addition, *S. agalactiae* also promotes invasive infections in non-pregnant adults, especially in patients with comorbidities such as liver dysfunction, cancer, obesity, and diabetes.[2] Sepsis caused by an aggressive inflammatory reaction of *S. agalactiae* can cause damage to multiple organ functions. Many bacteria produce virulence factors that can disrupt the endothelial barrier through a variety of mechanisms, including direct destruction of endothelial cells, alteration of the endothelial cytoskeleton, and disruption of cell-cell junctions during sepsis.

[3] Bacterial pilus plays a key role in immune activation and bacterial entry into the central nervous system.[4] In *S. agalactiae*, pilus type 2b (PI-2b) mediates adhesion and invasion of brain endothelial cells and contributes to translocation across the blood-brain barrier.[5] PI-2b also contributes to the pathogenesis of *S. agalactiae* infection by mediating invasion in several human epithelial cell lines (pulmonary A549, cervical HeLa, and colonic C2BBe1).[6,7] In addition, the presence of PI-2b increased phagocytosis of *S. agalactiae* by murine and human macrophages.[8] However, the pathway by which *S. agalactiae* crosses the endothelial barrier remains unclear.

Due to the luminal location, endothelial cells are specialised in detecting hemodynamic and mechanical changes, playing an essential role in maintaining blood fluidity, platelet aggregation and vascular tone. The important mechanical stimulus for endothelial cells is called wall shear stress, which is the frictional force ex-

Financial support: FAPERJ (Processo SEI 260003/005902/2024), Sub Reitoria de Pós-Graduação e Pesquisa da Universidade do Estado do Rio de Janeiro (SR-2/UERJ) provided graduate scholarships. This study was also financed in part by the CAPES (Finance Code 001 and Print/UERJ− award number 88881.311598/2018-01).
+ Corresponding author: pnagao@uol.com.br / pnagao@uerj.br
https://orcid.org/0000-0001-6007-0033

erted by blood flow applied tangentially to the endothelium. The most important mechanosensory complex is located at cell-cell junctions, which have been implicated as responsible for activating shear response pathways, including cell alignment and endothelial dysfunction.[9] Although the static culture model is easier and cheaper compared to the shear stress model, the expression of surface molecules, morphology, cell signalling, and interactions with different pathogens can occur heterogeneously between the two models. Both the cells and molecules that pass through the blood vessel and the endothelial cells that line the vessel suffer the action of this force, which is caused by the viscosity of the blood tangentially to these cells. Endothelial dysfunction or injury leads to disruption of the microvascular barrier, resulting in increased extravascular fluid, tissue oedema, and death in septic patients.[10]

Breakdown of VE-cadherin-mediated endothelial barrier function leads to altered vascular permeability and remodelling of endothelial cells, which are associated with several disease processes. Furthermore, fibrinogen promotes leukocyte transendothelial migration in a VE-cadherin-dependent manner,[11] and induces increased endothelial permeability to enable transendothelial cell migration.[12] Fibrinogen fragments/degradation products are present at high levels in the plasma after traumatic injury or infections and can circulate via the bloodstream to distant vascular beds. Fibrinogen fragments resulted in disruption of endothelial barrier integrity, which was associated with a decrease in endothelial cells expression of VE-cadherin and increased cell permeability.[13] In this work, we provide experimental evidence of the influence of shear stress during infection of HUVEC cells by *S. agalactiae*, as well as the role of PI-2b during endothelial barrier disruption and increased endothelial permeability.

## MATERIALS AND METHODS

*Bacterial strain origin and culture conditions* - *Streptococcus agalactiae* wild type (WT) GBS90356 (capsular type III, ST-17) and GBS90356Δ*pilus2b* (pilus 2b deficient mutant of GBS90356 kindly provided by Dr Kelly Doran - University of Colorado Anschutz School of Medicine) strains were used in this study. GBS90356 strain was the first Brazilian ST-17 strain sequenced in Brazil by our group and was partially investigated for virulence properties.[14,15,16,17,18] Both *S. agalactiae* strains were cultured on blood agar base plates containing 5% defibrinated sheep's blood (BAB; Sigma Aldrich, São Paulo, SP, Brasil) for 24 h at 37ºC. After, three colonies were grown in brain heart infusion broth (BHI; Sigma) at 37ºC until OD 540 reading of 0.4 [~$10^8$ colony forming units (CFU) per mL].[19]

*Human umbilical vein endothelial cells (HUVECs)* - HUVECs were obtained by treating umbilical veins with 0.1% collagenase IV (Sigma). HUVECs were seeded in 25 cm$^2$ flasks treated with porcine skin gelatin (Sigma) and cultured in medium 199 (M199 - Sigma) supplemented with 100 U mL$^{-1}$ penicillin, 100 µg mL$^{-1}$ streptomycin and 2.5 µg mL$^{-1}$ amphotericin-B, 2 mM glutamine and 20% foetal bovine serum (FBS) at 37ºC and 5% $CO_2$ until they reached confluence. Subsequently, confluent HUVECs monolayers (first or second passage) were treated with 0.025% trypsin/0.2% EDTA solution prepared in 0.01 M phosphate-buffered NaCl at pH 7.2 (PBS), rinsed in PBS, and used for experiments in culture plates (Corning, NY, USA).[19]

*Shear stress treatment* - HUVECs were submitted to steady laminar shear stress as previously described.[20] Briefly, HUVECs were seeded at 8 x $10^5$ cells/well in regular six-well plates and allowed to reach confluence, typically in two-three days. HUVECs were exposed to 10 dyn/cm$^2$ of constant shear stress for 24 h on an orbital rotator ($CO_2$ Resistant Shaker, Thermo Scientific™) using the equation $\tau w = \alpha\sqrt{\rho\eta}(2\pi f)^3$, where $\tau w$ is the shear stress, $\alpha$ is the radius of rotation (cm), $\rho$ is the density of the liquid (g/L), $\eta$ is fluid viscosity (0.0075 dyn/cm$^2$ at 37ºC), and f is the rotation per second. The flow device was kept at 37ºC for 24 h in a 5% $CO_2$ atmosphere. Each experimental condition was repeated three times. After shear stress, HUVECs monolayers were infected with *S. agalactiae* without orbital rotation as described below.

*Streptococcus agalactiae-HUVECs binding assays* - Confluent HUVECs cultured under static or shear stress conditions in tissue culture plates were infected with *S. agalactiae* (5 x $10^7$ CFU; MOI:100) in the presence or absence of fibrinogen (20 µg). Bacterial-cell contact was facilitated by centrifugation for 1 min at 200 g. Cells were then incubated at 37ºC in 5% $CO_2$ for 1 h. Infected HUVECs were rinsed with PBS, and lysed with 0.5 mL of 25 mM Tris, 5 mM EDTA, 150 mM NaCl, and 1% Igepal (lysis buffer). Bacterial recovery was determined by plating samples on BAB medium. Data were expressed as the mean adherent CFU mL$^{-1}$ of three experiments in triplicate.[19]

*Streptococcus agalactiae binding to immobilised fibrinogen* - Microtiter plates were coated overnight at 4ºC with 20 µg mL$^{-1}$ of fibrinogen (Sigma). The plates were washed three times with 0.5% (v/v) Tween 20 in PBS (PBST). To block additional protein-binding sites, wells were treated for 1 h at room temperature with 200 µL of 1% BSA in PBS. *S. agalactiae* (~ $10^9$ CFU 200 µL$^{-1}$) strains were added to triplicate wells of the microtiter plates and incubated for 1 h at room temperature. The plates were washed twice and stained with 0.5% (wt/vol) crystal violet. After, 50 mL of ethanol was added to each well to solubilise the dye. The OD at 550 nm was determined using a microplate reader. Each experiment was performed in triplicate, and the results were pooled and averaged.

*Immunofluorescence* - Endothelial cells infected with *S. agalactiae* strains (GBS90356WT or GBS90356Δ*pilus2b*) in the presence or absence of fibrinogen (20 µg) were stained with VE-cadherin mouse mAb (1:50; Santa Cruz, São Paulo, Brasil), and secondary Alexa fluor 488 donkey anti-mouse or secondary Alexa fluor 647 goat anti-mouse (Santa Cruz) during 1 h at 37ºC. Coverslips were mounted on slides with fluorescent mounting medium containing 40,6-diamidino-2-phenylindole for staining of nuclei (ProLong; Invitrogen). Images were acquired

with an inverted epifluorescence microscope Olympus IX71 TH4-100 model. VE-cadherin fluorescence intensity was quantified using the free image analysis software ImageJ (https://imagej.net/ij/index.html). For phalloidin staining, HUVECs were washed with PBS, fixed for 15 min with 4% paraformaldehyde at 4ºC, followed by incubation in 0.5% Triton X-100 solution for 5 min. The cytoskeleton was stained with phalloidin in the dark at room temperature for 30 min. The fluorescence images were observed under a fluorescence microscope.

*Isolation of RNA and reverse transcriptase polymerase chain reaction (RT-PCR)* - HUVECs were collected by a cell-scraper after *S. agalactiae* infection, and total RNA was isolated using TRIzol reagent (Invitrogen, São Paulo, SP, Brazil) according to the manufacturer's protocol. The purity (A260/A280) and concentration of RNA were determined using a NanoDrop 2000 spectrophotometer (Thermo Fisher Scientific, São Paulo, SP, Brazil). Using the Maxima First Strand cDNA Synthesis Kit with dsDNase (Thermo Fisher Scientific), reverse transcription of cDNA was carried out in accordance with the manufacturer's instructions. Real-time reverse transcription PCR (RT-PCR) amplification was performed with SYBR Green Master Mix in a StepOne PCR amplifier (Thermo Fisher Scientific). Untreated HUVECs were used as the reference sample and β-actin was used as the endogenous control. The PCR primer sequences and annealing temperatures for PCR were: VE-cadherin (230 bp, Tm 55°C and 30 cycles) Forward 5'ACATCACAGTCAAGTATG-GGC3' Reverse 5'GATGCAGAGTAAGATGGCTGC 3' and β-actin (289 bp, Tm 58°C and 25 cycles) Forward 5' TGGACTTCGAGCAAGAGATGG3' Reverse 5'ATCTCTTCTGCATCCTGTCG3' as described before.[21] The amplified DNA products were separated on 2% agarose gel, stained with ethidium bromide, visualised and photographed. The assay was performed in triplicate, and each experiment was repeated at least three times.

*Permeability assay* - Sheared or static endothelial cells were plated at a density of $4 \times 10^5$ cells mL$^{-1}$ on the upper chamber of hanging inserts (Millicell, pore size of 0.4 μM; Millipore), and endothelial cell medium was added to the lower chamber. Following infection, FITC–dextran (250 μg mL$^{-1}$; 40 kDa) was added to the endothelial cells in the upper chamber. Permeability was determined after 24 h by measuring the amount of FITC-dextran that permeated through the endothelial cells into the lower chamber with a fluorescent plate reader (1420 Victor V3; Perkin Elmer). The positive control corresponded to the translocation of FITC-dextran in the absence of HUVECs and bacteria. The negative control corresponded to monolayers of endothelial cells with intact adherent junctions, without bacteria. Data are expressed as percentages of 100% permeability.[22]

*Statistical analysis* - Data are presented as mean value and standard error of the mean (SEM) of at least three independent experiments. Differences between groups were tested using Student's t-test, in the case of bound groups; the paired t-test was applied. Differences were considered to be significant at $p < 0.05$.

*Ethics* - This project was approved by the Ethics Committee of the State Secretariat of Rio de Janeiro, receiving the number CAAE 51317415.8.3001.5259.

## RESULTS

*Shear stress induces the alignment of HUVECs* - Phalloidin staining was performed to evaluate cell morphology of HUVECs in static and shear stress conditions. HUVECs in static model revealed a cobblestone morphology with no preferred orientation (Fig. 1A), whereas cells cultured under shear stress (10 dyn/cm$^2$) showed a higher degree of cellular alignment along the direction of the flow (Fig. 1B).

*Increased adherence of S. agalactiae strains on HUVEC cultured under shear stress* - *Streptococcus agalactiae* strains were able to adhere to HUVEC monolayers cultured under static or shear stress conditions. The GBS90356WT strain showed higher adhesion capacity on HUVECs cultured under shear stress when compared to HUVECs in a static model after 1 h post-infection ($p < 0.001$; Fig. 1C). Similar results were observed for GBS90356*Δpilus2b* ($p < 0.02$; Fig. 1C). Furthermore, the GBS90356WT strain demonstrated higher adhesion in both models static ($p < 0.01$) and shear stress ($p < 0.001$) when compared to the PI-2b mutant strain (Fig. 1C).

*Streptococcus agalactiae disrupts VE-cadherin junctions and increases endothelial permeability* - Immunofluorescence analysis showed a higher peripheral (membrane) localisation of VE-cadherin in HUVECs under shear stress compared to HUVECs cultured in static conditions (Fig. 2Aa-b). However, 1 h post-infection with bacterial strains, VE-cadherin labelling presented a diffuse and discontinuous pattern (Fig. 2Ac-f) with rupture points, showing a clear pattern of disruption of cell-cell junctions, and damage to the integrity of the endothelial monolayer under shear stress (Fig. 2Ad-f), mainly with the *S. agalactiae* GBS90356WT strain (Fig. 2Ad). Using imageJ software, we verified an increased VE-cadherin fluorescence in HUVEC exposure to shear stress (Fig. 2B; $p < 0.001$). The results were supported by RT-PCR analyses that identified significant increases in VE-cadherin expression in HUVECs subjected to shear stress compared to static HUVECs (Fig. 2B; $p < 0.001$). In contrast, both *S. agalactiae* strains induced a decrease of VE-cadherin expression in HUVECs under shear stress (Fig. 2B), mostly for GBS90356WT strain (Fig. 2B; $p < 0.01$). The results were consistent with endothelial barrier disruption, which the permeability levels were significantly increased in infected HUVECs compared to controls (Fig. 2D; $p < 0.001$).

*Fibrinogen promotes binding of S. agalactiae strains to HUVECs* - Both *S. agalactiae* strains (GBS90356WT and GBS90356*Δpilus2b*) were able to adhere to immobilised fibrinogen, where the mutant showed lower potential when compared to the wild type strain (Fig. 3A; $p < 0.001$). To evaluate whether fibrinogen influences bacterial adhesion to HUVECs in response to endothelial cell growth conditions, bacterial-cell interaction was evaluated using HUVECs grown in static or under shear stress condition. In the presence of fibrinogen, the ad-

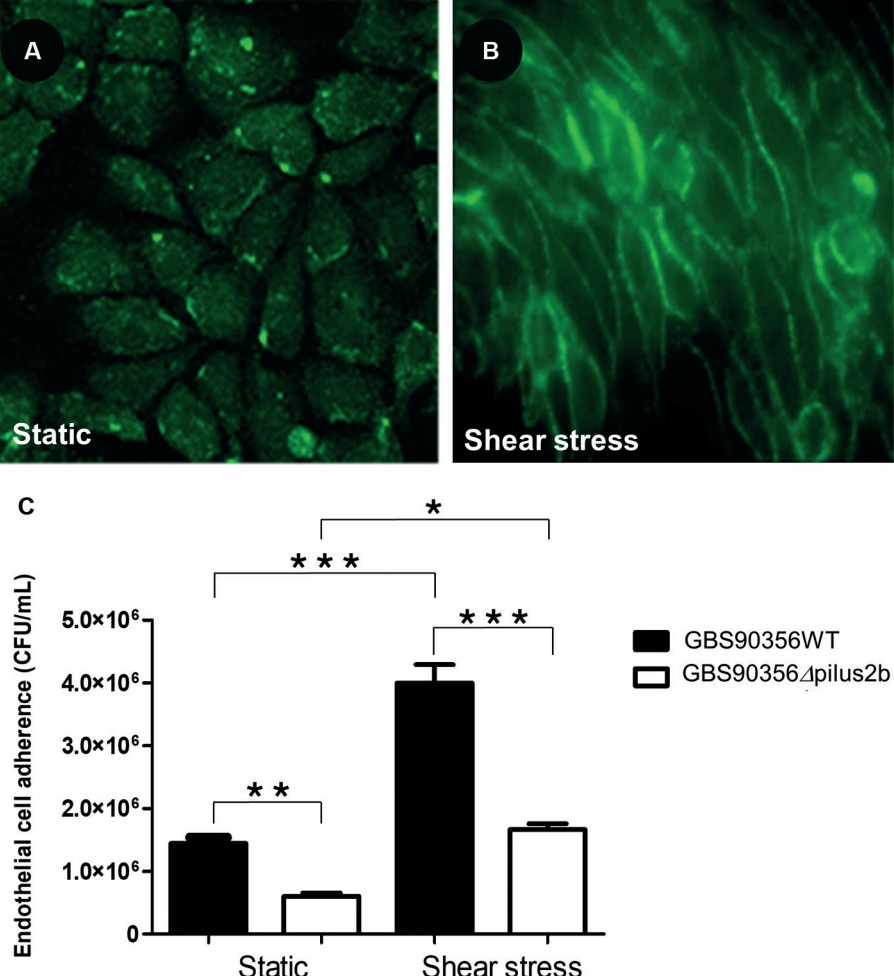

Fig. 1: cell morphology of human umbilical vein endothelial cells (HUVECs). F-actin fluorescence analysis on HUVECs cultured in static (A) and *shear* stress (B) conditions. (C) Adherence of *Streptococcus agalactiae* (GBS9056WT and GBS90356Δpilus2b) on HUVECs under static and shear stress conditions. Results are presented as means ± standard error of the mean (SEM) from at least three independent experiments in triplicate wells. Asterisk indicates *p < 0.02, **p < 0.01, ***p < 0.001.

hesive capacity of the bacterial strains was increased, especially in HUVECs cultured under shear stress. Furthermore, the GBS90356WT strain showed the highest adhesion in all conditions, especially in the presence of fibrinogen (Fig. 3B; p < 0.001). VE-cadherin staining showed several rupture points and numerous intercell spaces, demonstrating the participation of fibrinogen in the disruption of adherent's junctions, especially in HUVECs under shear stress infected by GBS90356WT strain (Fig. 3C; p < 0.001).

## DISCUSSION

The endothelium plays a key role during the development and progression of sepsis. Loss of the endothelial barrier can lead to increased leukocyte adhesion and transmigration, influx of large molecules and water across the endothelium, which can lead to multiple organ failure.[23] Endothelial cells are exposed to shear stress, a fundamental physical characteristic of endothelial cell alignment, critical for vascular homeostasis.[24] Our results confirmed that shear stress at physiological mag-

nitudes is required for HUVECs to adopt an elongated and axially polarised phenotype in the flow direction, important for maintaining a functionally confluent cell monolayer. Based on this information, we performed the interaction of *S. agalactiae* strains with HUVECs under both static and shear stress growth conditions.

*Streptococcus agalactiae* ST-17 strains have been particularly well studied given their role as the major cause of neonatal disease, as well as the contribution of PI-2b to the virulence of ST-17 strains. PI-2b expression is regulated in GBS ST-17 strains by a 43-bp hairpin-like structure in the upstream region of PI-2b operon, conferring a selective advantage in the human host, either by reducing host immune responses or by increasing its dissemination potential. The PI-2b locus is found primarily in *S. agalactiae* type III ST-17, but may also be present in non-ST17 strains isolated from humans.[25] Previous results suggested that high expression of the PI-2b locus in *S. agalactiae* strains could contribute to biofilm formation. The authors suggested that a complex regulation of PI-2b expression, indirectly mediated by CovR and

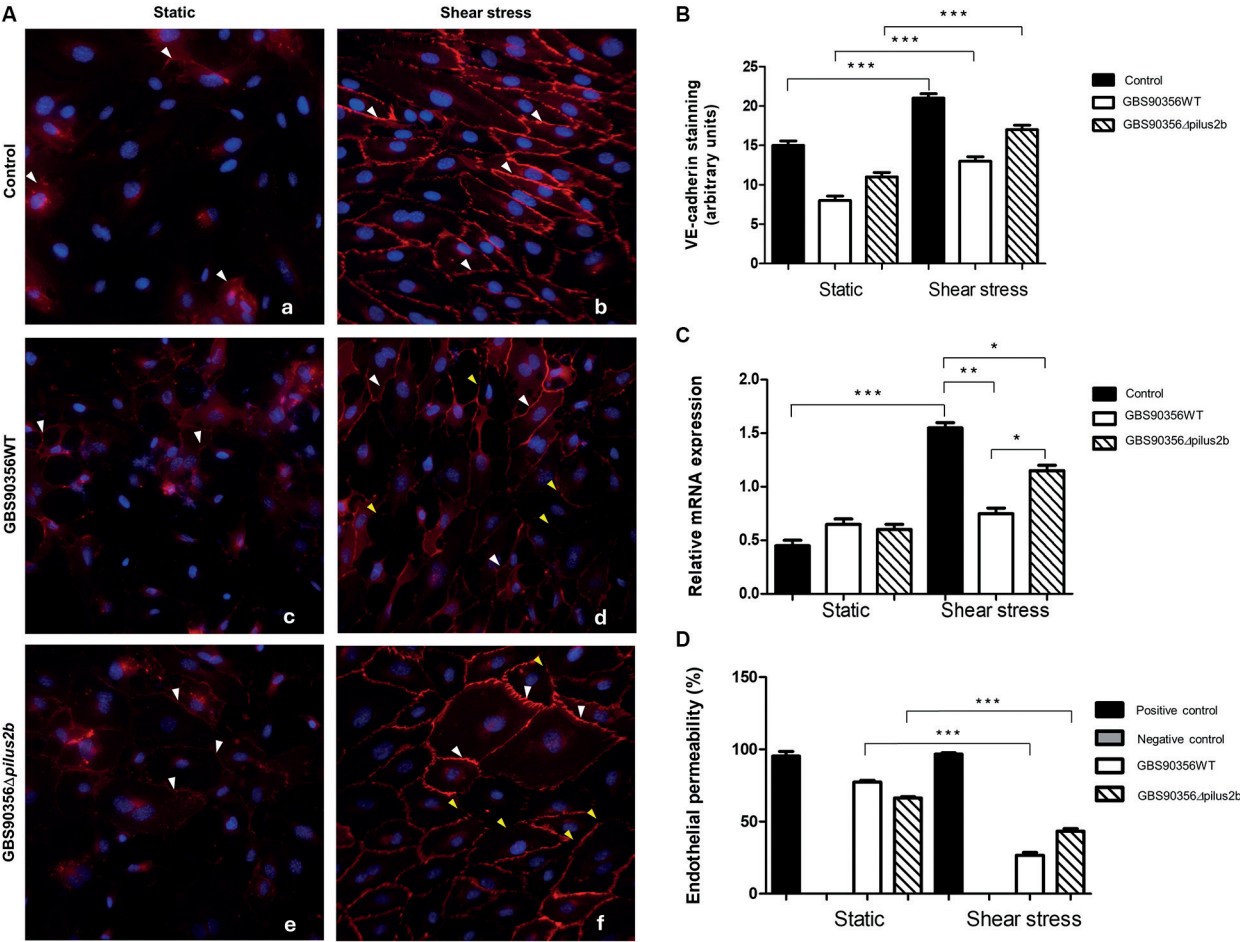

Fig. 2: *Streptococcus agalactiae*-induced disturbances of VE-cadherin-mediated cell-cell interactions. (A) Immunofluorescent staining for VE-cadherin (red) and nuclei (blue) in human umbilical vein endothelial cells (HUVECs) under static (Aa, c, e) or shear stress conditions (Ab, d, f). HUVECs monolayer was stimulated with GBS9056WT (Ac, d) or GBS90356Δpilus2b (Ae, f) strains. (B) Quantification of VE-cadherin fluorescence intensity using ImageJ image analysis software. (C) mRNA expression in HUVECs were evaluated using real-time polymerase chain reaction (RT-PCR) after stimulation with *S. agalactiae* for 1 h. (D) HUVECs monolayer permeability was evaluated using the Millicell system. White arrowheads show intact VE-cadherin barrier, yellow arrowheads show VE-cadherin barrier disruption. Values are presented as means ± standard error of the mean (SEM) (n = 3) (*p < 0.02, **p < 0.01, ***p < 0.001).

other *S. agalactiae*-specific regulatory factors, would be involved in this process.[26] Subsequently, the same authors showed that higher expression of the PI-2b pilus polymer resulted in increased phagocytosis by human monocyte-derived THP-1 macrophages.[7] Moreover, the role of PI-2b in ST-17 pathogenesis by enhancing cell invasion, bacteraemia and spread into the central nervous system, suggested that this protein could be a therapeutic and prophylactic target against neonatal sepsis and meningitis caused by *S. agalactiae*.[5]

In the present work, there was a greater adhesive capacity of both *S. agalactiae* strains (GBS90356WT and GBS90356Δpilus2b) in HUVECs cultured in the shear stress than in the static condition. In addition, the GBS90356WT strain presented a higher viable bacteria count (CFU mL⁻¹) when compared to the mutant strain for PI-2b, confirming the participation of this molecule during endothelial infection by this pathogen. *S. agalactiae* strains isolated from neonatal invasive infections in European countries showed high expression of PI-2b compared with PI-1.[27] Expression of the PI-2b gene was two-fold higher in early-onset disease isolates compared to colonising isolates and mainly detected in capsular type III.[28] The contribution of PI-2b to the virulence of ST-17 strains enhanced the efficiency of non-opsonic macrophage phagocytosis and conferred a survival advantage inside macrophages, promoting phagocyte resistance and systemic virulence.[29] PI-2b also contributed to the initial attachment and invasion of *S. agalactiae* to cervical ME-180 and alveolar epithelial A549 cells.[30] In agreement, our results demonstrated that the mutant *S. agalactiae* strain presented less adhesive capacity on HUVECs and less ability to disrupt junctions, when compared with GBS90356WT, showing the contribution of PI-2b to dissemination in the endothelial host cells.

Located in the basolateral regions of cells, the adherent junctions are responsible for maintaining cellular integrity. Among the cadherins, we can highlight the fundamental role of vascular endothelial cadherin (VE-cadherin), which directly regulates endothelial

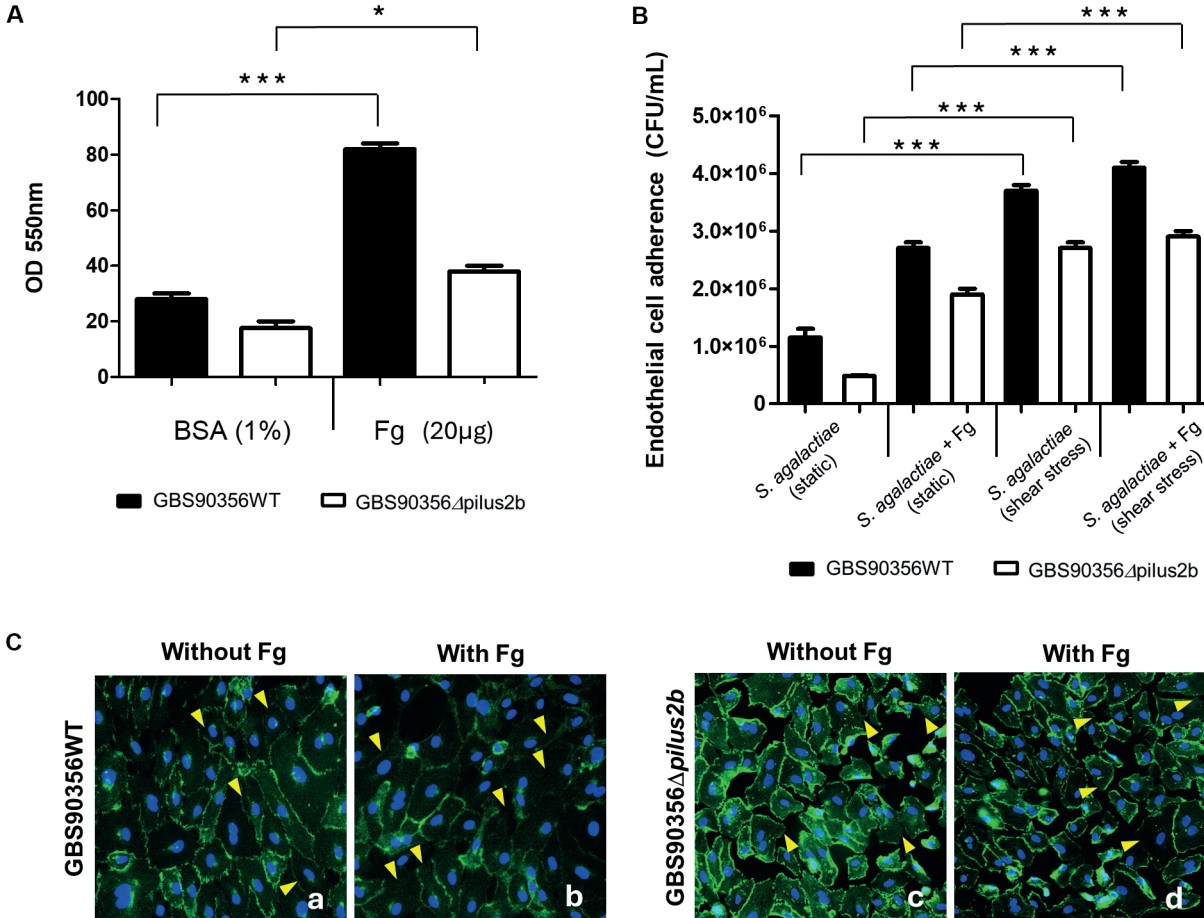

Fig. 3: effect of fibrinogen on *Streptococcus agalactiae*-endothelial cell interaction. (A) *S. agalactiae* (GBS9056WT and GBS90356Δpilus2b) binding to immobilised fibrinogen (20 μg). (B) Fibrinogen enhances adherence of *S. agalactiae* in both human umbilical vein endothelial cells (HUVECs) grown in static or shear stress conditions. (C) Immunofluorescent staining for VE-cadherin (green) and nuclei (blue) in HUVECs under shear stress stimulated with GBS9056WT or GBS90356Δpilus2b in the presence of fibrinogen. White arrowheads show intact VE-cadherin barrier, yellow arrowheads show VE-cadherin barrier disruption. Data represent the mean ± standard error of the mean (SEM); *p < 0.02, ***p < 0.001.

permeability and is also capable of binding to plasma proteins, increasing the capacity of the endothelium to respond to different stimuli from the extracellular environment.[31] In our study, VE-cadherin upregulation was verified in HUVEC under shear stress and downregulation after infection by *S. agalactiae* strains, mainly for GBS90356WT. The decrease in VE-cadherin expression by the GBS90356Δpilus2b strain was lower when compared to the GBS90356WT strain, demonstrating the role of PI-2b during disease progression.

A previous study demonstrated that adherent's junction VE-cadherin controls tight junction organisation by promoting increased permeability. This mechanism occurs through upregulation of tight junction claudin-5, mediated by downregulation of VE-cadherin.[32] The integrity of the blood-brain barrier was also increased after increased expression of claudin-5 and VE-cadherin in neurons.[33] The link between adherents junction and tight junctions explains why VE-cadherin inhibition can cause a marked increase in permeability. Thus, we can suggest that VE-cadherin downregulation by *S. agalactiae* strains is an important pathogenic mechanism to increase endothelial permeability and promote bacterial dissemination during sepsis. In addition, both *S. agalactiae* strains (GBS90356WT and GBS90356Δpilus2b) caused VE-cadherin disruption, further increasing the invasive capacity of *S. agalactiae*, especially in the presence of PI-2b.

Some publications emphasise the importance of fibrinogen in the adhesion processes of *S. agalactiae* to host cells, including the surface proteins as FbsA, FbsB and FbsC,[34] Srr1 and Srr2.[2] In this work, the presence of *fbsA, fbsB, Srr1* and *Srr2* genes were detected in GBS90356 strain (data not shown). Consistently, *S. agalactiae* ST-17 clinical isolates exhibit a higher capacity to bind fibrinogen when compared with non-ST-17 strains.[35] Srr2 has a higher affinity for fibrinogen than Srr1, suggesting that its expression may be critical for the hypervirulence of ST-17 isolates. Adhesion to fibrinogen is essential for early colonisation of host tissues and organs. Invading bacterial pathogens frequently bind to plasminogen/plasmin, and the resulting increase in cell surface proteolytic activity contributes to their subsequent dissemination within the infected host by facilitating the crossing of barriers such as the blood-brain barrier.[36] Moreover, few studies reported the association between pili and fibrinogen. The

contributions of the individual pilus subunits EmpA and EmpB during the adhesion of *Enterococcus faecium* to fibrinogen and type I collagen was demonstrated.[37] Interestingly, significant reductions in adherence to fibrinogen and collagen type I were observed with deletion of *emp*A and *emp*B genes when compared to the wild type. Our results corroborate the previous data, where the absence of the pilus subunit (PI-2b) significantly reduced the binding of the *S. agalactiae* strains to fibrinogen, as well as the adhesive capacity on HUVECs, predominantly under shear stress conditions.

Although reductions in VE-cadherin expression and vascular barrier integrity following trauma and haemorrhage have been described, the mechanisms remain unknown.[38] Various coagulation factors, including fibrinogen and/or fibrinogen degradation products, modulate the inflammatory response by affecting leukocyte migration and cytokine production.[39] Vascular endothelial barrier disruption is a central mechanism in the development of multiple organ failure after severe trauma and haemorrhage. Fibrinolysis occurs during organ dysfunction, resulting in high circulating levels of fibrinogen degradation products, such as fragment X that induces endothelial cell hyperpermeability by reducing VE-cadherin expression. In addition to affecting VE-cadherin, fragment X also induced significant changes in genes that regulate endothelial cell-mediated coagulation, inflammation, angiogenesis, and vasoconstriction.[40] Thus, fibrinogen and/or its fragments increased the severity of several inflammatory conditions. Presently, our results could suggest that *S. agalactiae* uses fibrinogen binding to induce changes in the expression of paracellular junctional proteins that regulate endothelial cell barrier function.

In conclusion, the fine-tuning PI-2b expression *in vivo* is essential to optimise *S. agalactiae* adherence to fibrinogen/fragment products and endothelial cells, favouring the breakdown of adherent junctions through VE-cadherin, and providing increased endothelial permeability for microorganism dissemination.

### AUTHORS' CONTRIBUTION

JSSO - Formal analysis, investigation, methodology, validation, writing - original draft; BASPH - formal analysis, investigation, methodology, validation; LNF - investigation, methodology, visualisation; MENS - investigation, methodology, visualization; GSS - formal analysis, investigation, resources, writing - review & editing; PEN - conceptualisation, formal analysis, funding acquisition, project administration, supervision, visualisation, writing - review & editing. The authors declare that the research was conducted in the absence of any commercial or financial relationships that could be construed as a potential conflict of interest.

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

# OPEN PEER REVIEW

Memórias do IOC thanks the anonymous reviewers for their contribution to the peer review of this work.

## FIRST REVIEW ROUND

REVIEWERS' COMMENTS

### REVIEWER #1

a) The abstract is informative;

b) The work presented in this manuscript is a good contribution to the streptococcus agalactiae field. It provides further mechanisms for the field to understand the ability of the organism to adhere and disrupt the endothelial barrier;

c) Methodology:

- It would be useful for the authors to provide more details regarding the sheer stress experimental set up. Explaining what manufacturers system was used to maintain the constant flow, how long the cells were under the sheer stress before imaging, how were the cells infected in the system, etc.

- In the statistic section it is stated that the figures show mean and SEM but all the figure legends state mean and SD. Can the authors clarify this.

d) References are fine;

e) Figures:

- In Figure 2 it would be beneficial to increase the "a-f" lettering in the microscope images as these are very difficult to see currently.

- In each figure legend there is a typo for the GBS WT strain (9056WT instead of 90356WT).

### REVIEWER #2

In their study, the authors demonstrate that the hypervirulent Group B Streptococcus strain GBS90356 exhibits enhanced adherence (a 4-5 fold increase) to human vein endothelial cells (HUVEC) under shear stress conditions. This adherence is reduced when using a PI-2b pilus mutant. The findings also suggest that Group B Streptococcus can disrupt cell junctions by decreasing VE-cadherin expression under shear stress conditions and that the addition of external fibrinogen further enhances GBS adherence to endothelial cells. However, the overall rationale of the manuscript could be clearer, and the experimental model using HUVEC, as well as the results presented in Figures 2 and 3, may benefit from additional validation and explanation.

The title of the manuscript is misleading and not backed up by the results.

Main Comments:

1. Strain Selection: The authors focus on a single GBS strain, GBS90356, which was isolated from a meningitis case and is not well-characterized. To strengthen their findings, it would be beneficial to include additional strains, such as another CC-17 strain like COH1 and its isogenic COH1ΔPI2b mutant, for comparison.

2. Rationale for PI2b Mutant: The choice to test the PI2b mutant should be better explained. Previous studies by Lazzarin et al. and Perichon et al. have shown the role of PI2b in adhesion to epithelial alveolar cells (A549) and human brain microvascular endothelial cells (hBMEC). These findings could provide a stronger context for the current study.

3. Fibrinogen Adherence: It is well-established that CC-17 strains adhere better to fibrinogen, primarily due to the surface expression of the Srr2 glycoprotein (Six et al., Molecular Microbiology 2015). Including this reference in the discussion would enrich the manuscript.

4. VE-Cadherin Downregulation: The immunofluorescence images showing VE-cadherin downregulation upon GBS infection could be more convincing. Including other proteins that constitute cell junctions for analysis would provide a more comprehensive view. Additionally, a study by Campeau et al. (2020) performed proteome profiling of blood-brain barrier perturbation by GBS and did not report major alterations of endothelial junctions in vivo. This study should be considered in the discussion.

5. Transcriptional Repressor SNAIL-1: The authors should also analyse the possible upregulation of the transcriptional repressor SNAIL-1, as previously reported by Kim et al., J Clin Invest. 2015 leading to the repression of tight junction genes and blood-brain barrier disruption. This would add depth to the interpretation of their findings.

6. Figure 2C: Fig. 2C requires clarification. The authors should explain that the positive control (100%) corresponds to FITC dextran translocation in the absence of cells and bacteria. A negative control (0%) should be included, showing the presence of endothelial cells forming tight junctions without bacteria. The effect of adding bacteria to the filter should also be demonstrated.

7. Figure 3: The relevance of the results shown in Fig. 3 is unclear. If fibrinogen is important for leukocyte transendothelial migration in a VE-cadherin-dependent manner, the authors should explain their conclusions from

this experiment and from the previous one showing that GBS disrupt the VE-cadherin. Additionally, they should justify the use of 20 μg of fibrinogen and compare this concentration with 1% BSA.

**REVIEWER #3**

Overall, the manuscript is well-written and the content contributes to the knowledge of GBS pathogenesis. I have no special concern of any section of the manuscript. Below, a list of recommendations and suggestions.

Page 4:

L 16: "stillbirths deaths" is redundant. Are authors supposed to mention "stillbirths and infant deaths"? In addition, it would be valuable to include a reference for these epidemiological data.

L 25-27: "virulence factors that can disrupt the endothelial barrier, which can directly kill endothelial cells and/or disrupt endothelial cell-cell junctions during sepsis"… it is somewhat repetitive. I suggest improving this sentence.

L57: "morphology, cell signaling,…" the word "AND" is needed after "signaling".

Page 8:

L41: Please correct: MILLICELL

Page 11:

L30-32: "characteristic of endothelial cell alignment that is critical for vascular homeostasis..." The sentence is too large and can be shortened. I suggest removing "that is".

L45: "severe invasive infant disease..." I suggest changing to "neonatal disease".

L53: "advantage in the human host by reducing host immune responses…" The sentence is too large and can be shortened. I suggest that the authors revise and improve it.

L57: "but may also be present in non-ST17 human strains..." it is somewhat strange "human strains". I suggest that the authors revise and improve it.

Page 12:

L37: I suggest removing "Epidemiological." Despite the large sample from different countries, the study has focused on investigating antibodies against GBS cps and pilus.

Page 13:

L25-27: "Decreased expression of VE-cadherin by the GBS90356Δpilus2b strain was less than the reduction caused by the GBS90356WT strain, demonstrating the involvement of PI-2b". I suggest the authors improve this sentence, especially the end, "involvement of PI-2b".

L29-30: I suggest the authors improve the sentence. They should initiate this paragraph by clarifying that the text is a result of other studies, since "The authors" appears in the middle of the paragraph, but the reader has no idea until he reaches this point. Another alternative is simply remove "The authors", reorganize the sentence, and reference it.

L41: "explaining" should be changed to "explains".

Page 14:

L37-39: "Previous data demonstrated that fibrinogen resulted in disruption of endothelial barrier integrity…" I recommend the authors to improve the sentence: the original reference mentions fibrinogen fragments, degradation products of fibrinogen, not fibrinogen itself. So, the subsequent discussion should be carefully revised.

EDITOR COMMENTS

In addition to responding to all points raised by reviewers, it is necessary to:

(i) improve the methodology sections with more details of some experiments performed as suggested by the reviewers. In addition, in the section of RT-PCR: "cDNA synthesis and PCR were performed as described previously." There is no reference, and the authors need to briefly describe quantities and conditions used in the reactions.

(ii) The discussion should be carefully revised with the reviewers' suggestions, including more articles from the literature.

(iii) Page 15:  "In this work, the presence of fbsA, fbsB, Srr1 and Srr2 genes were detected in GBS90356 strain (data not shown)." It would be important to include this data as a Supplementary Figure of the manuscript.

**AUTHORS' RESPONSE TO THE REVIEWERS**

Dr. Geovane Dias-Lopes
Handling Editor
Memórias do Instituto Oswaldo Cruz

Dear Dr. Dias-Lopes,

Thank you for your message and critical comments concerning the manuscript ID MIOC-2025-0077 entitled "Streptococcus agalactiae disrupts VE-cadherin intercellular junctions and induces endothelial barrier dysfunction"" by Oliveira and co-workers. Following the suggestions made in the first version submitted to Memórias do

Instituto Oswaldo Cruz, we are submitting the revised version. The manuscript was revised and supplied with new information to improve data quality, as pointed out by the reviewers.

Sincerely,

Professor Dr Prescilla Emy Nagao

## REVIEWERS' COMMENTS

### REVIEWER #1

a) The abstract is informative.

b) The work presented in this manuscript is a good contribution to the Streptococcus agalactiae field. It provides further mechanisms for the field to understand the ability of the organism to adhere and disrupt the endothelial barrier.

c) Methodology:

- It would be useful for the authors to provide more details regarding the sheer stress experimental set up. Explaining what manufacturers system was used to maintain the constant flow, how long the cells were under the sheer stress before imaging, how were the cells infected in the system, etc.

Page 7, lines 143-152: HUVECs were submitted to steady laminar shear stress as previously described (Colgan et al., 2007). Briefly, HUVECs were seeded at 8x105 cells/well in regular six-well plates and allowed to reach confluence, typically in 2-3 days. HUVECs were exposed to 10 dyn/cm² of constant shear stress for 24 h in orbital rotator (CO2 Resistant Shaker, Thermo Scientific™) using the equation tw = α√rh(2pf)3, where tw is the shear stress, α is the radius of rotation (cm), r is the density of the liquid (g/L), h is fluid viscosity (0.0075 dyn/cm² at 37 °C), and f is the rotation per second. The flow device was kept at 37°C for 24 h in a 5% CO2 atmosphere. Each experimental condition was repeated three times. After shear stress, HUVECs monolayers were infected with S. agalactiae without orbital rotation as described below.

- In the statistic section it is stated that the figures show mean and SEM but all the figure legends state mean and SD. Can the authors clarify this.

Answer: There was a typing error. The results were expressed as "Mean and Standard Error of the Mean (SEM) and not Standard Deviation (SD). The figure legends have been corrected.

d) References are fine.

e) Figures:

- In Figure 2 it would be beneficial to increase the "a-f" lettering in the microscope images as these are very difficult to see currently.

- In each figure legend there is a typo for the GBS WT strain (9056WT instead of 90356WT).

Answer: Figure 2 has been corrected.

### REVIEWER #2

In their study, the authors demonstrate that the hypervirulent Group B Streptococcus strain GBS90356 exhibits enhanced adherence (a 4-5 fold increase) to human vein endothelial cells (HUVEC) under shear stress conditions. This adherence is reduced when using a PI-2b pilus mutant. The findings also suggest that Group B Streptococcus can disrupt cell junctions by decreasing VE-cadherin expression under shear stress conditions and that the addition of external fibrinogen further enhances GBS adherence to endothelial cells. However, the overall rationale of the manuscript could be clearer, and the experimental model using HUVEC, as well as the results presented in Figures 2 and 3, may benefit from additional validation and explanation.

The title of the manuscript is misleading and not backed up by the results.

Answer: Title was changed to "Infection of endothelial cells by Streptococcus agalactiae reveals potential role of PI-2b pilus on endothelial barrier dysfunction".

Main Comments:

1. Strain Selection: The authors focus on a single GBS strain, GBS90356, which was isolated from a meningitis case and is not well-characterized. To strengthen their findings, it would be beneficial to include additional strains, such as another CC-17 strain like COH1 and its isogenic COH1ΔPI2b mutant, for comparison.

Answer: Strain GBS90356 is well characterized and has its whole genome sequenced and published by our group as the first Brazilian ST-17 strain sequenced (Lannes-Costa PS, Baraúna RA, Ramos JN, Veras JFC, Conceição MVR, Vieira VV, de Mattos-Guaraldi AL, Ramos RTJ, Doran KS, Silva A, Nagao PE. Comparative genomic analysis and identification of pathogenicity islands of hypervirulent ST-17 Streptococcus agalactiae Brazilian strain. Infect Genet Evol. 2020 Jun;80:104195. doi: 10.1016/j.meegid.2020.104195). GBS90356 strain was also studied for virulence properties (Costa et al. 2011; Oliveira et al., 2018; Lannes-Costa et al., 2020, da Conceição et al., 2023; Lannes-Costa et al., 2024). The reviewer's suggestion to include COH1 strain and its isogenic mutant COH1ΔPI2b for comparison is not relevant in this experimental assay. Published studies evaluating the importance

of pilus 2b (or other molecules such as SaeRS, LytSR, BvaP, SfpB) in the virulence of S. agalactiae did not use more than one wild-type strain and its isogenic mutant belonging to ST-17 or another capsular type of a clinical isolate of S. agalactiae (some publications listed below). This is currently justified because the comparisons involve other factors and variables specific to each strain belonging to the same clone (already demonstrated by proteomics and metabolomics), which can significantly interfere in the final analysis of the results obtained. Our work highlights the importance of analyzing the virulence profile of the first Brazilian ST-17 strain sequenced in Brazil (GBS90356), whose data demonstrate for the first time the pathogenic potential of dissemination and disruption of adherens junctions by S. agalactiae, mainly in endothelial cells under physiological shear stress.

(i) Lazzarin et al., 2017. Contribution of pilus type 2b to invasive disease caused by a Streptococcus agalactiae ST-17 strain. BMC Microbiol. 3;17(1):148. doi: 10.1186/s12866-017-1057-8;

(ii) Campeau A, Mills RH, Blanchette M, Bajc K, Malfavon M, Munji RN, Deng L, Hancock B, Patras KA, Olson J, Nizet V, Daneman R, Doran K, Gonzalez DJ. Multidimensional Proteome Profiling of Blood-Brain Barrier Perturbation by Group B Streptococcus. mSystems. 2020 Aug 25;5(4):e00368-20. doi: 10.1128/mSystems.00368-20;

(iii) Thomas LS, Cook LC. A Novel Conserved Protein in Streptococcus agalactiae, BvaP, Is Important for Vaginal Colonization and Biofilm Formation. mSphere. 2022 Dec 21;7(6):e0042122. doi: 10.1128/msphere.00421-22;

(iv) Li S, Li W, Liang Q, Cao J, Li H, Li Z, Li A. Characterization and virulence of Streptococcus agalactiae deficient in SaeRS of the two-component system. Front Microbiol. 2023 Apr 17;14:1121621. doi: 10.3389/fmicb.2023.1121621;

(v) AlQadeeb H, Baltazar M, Cazares A, Poonpanichakul T, Kjos M, French N, Kadioglu A, O'Brien M. The Streptococcus agalactiae LytSR two-component regulatory system promotes vaginal colonization and virulence in vivo. Microbiol Spectr. 2024 Nov 5;12(11):e0197024. doi: 10.1128/spectrum.01970-24;

(vi) Li H, Cao J, Han Q, Li Z, Zhuang J, Wang C, Wang H, Luo Z, Wang B, Li A. Protease SfpB plays an important role in cell membrane stability and immune system evasion in Streptococcus agalactiae. Microb Pathog. 2024 Jul;192:106683. doi: 10.1016/j.micpath.2024.106683.

Page 6, Lines 124-125: GBS90356 strain was the first Brazilian ST-17 strain sequenced in Brazil by our group and was partially investigated for virulence properties (Costa et al. 2011; Oliveira et al., 2018; Lannes-Costa et al., 2020, da Conceição et al., 2023; Lannes-Costa et al., 2024).

2. Rationale for PI2b Mutant: The choice to test the PI2b mutant should be better explained. Previous studies by Lazzarin et al. and Perichon et al. have shown the role of PI2b in adhesion to epithelial alveolar cells (A549) and human brain microvascular endothelial cells (hBMEC). These findings could provide a stronger context for the current study.

Page 4, Lines 83-90: In S. agalactiae, pilus type 2b (PI-2b) mediates adhesion and invasion of brain endothelial cells and contributes to translocation across the blood-brain barrier (Lazzarin et al., 2017). PI-2b also contributes to the pathogenesis of S. agalactiae infection by mediating invasion in several human epithelial cell lines (pulmonary A549, cervical HeLa, and colonic C2BBe1) (Adderson et al., 2003; Périchon et al., 2019). In addition, the presence of PI-2b increased phagocytosis of S. agalactiae by murine and human macrophages (Chattopadhyay et al. 2011). However, the pathway by which S. agalactiae crosses the endothelial barrier remains unclear.

3. Fibrinogen Adherence: It is well-established that CC-17 strains adhere better to fibrinogen, primarily due to the surface expression of the Srr2 glycoprotein (Six et al., Molecular Microbiology 2015). Including this reference in the discussion would enrich the manuscript.

Answer: The reference Six et al., Molecular Microbiology, 2015 was included in the manuscript discussion.

Page 15, Lines 346-353: Consistently, S. agalactiae ST-17 clinical isolates exhibit a higher capacity to bind fibrinogen when compared with non-ST-17 strains (Dramsi et al., 2012). Srr2 has a higher affinity for fibrinogen than Srr1, suggesting that its expression may be critical for the hypervirulence of ST-17 isolates. Adhesion to fibrinogen is essential for early colonization of host tissues and organs. Invading bacterial pathogens frequently bind to plasminogen/plasmin, and the resulting increase in cell surface proteolytic activity contributes to their subsequent dissemination within the infected host by facilitating the crossing of barriers such as the blood-brain barrier (Six et al., 2015).

4. VE-Cadherin Downregulation: The immunofluorescence images showing VE-cadherin downregulation upon GBS infection could be more convincing. Including other proteins that constitute cell junctions for analysis would provide a more comprehensive view. Additionally, a study by Campeau et al. (2020) performed proteome profiling of blood-brain barrier perturbation by GBS and did not report major alterations of endothelial junctions in vivo. This study should be considered in the discussion.

Answer: Campeau et al. (2020) compared the COH1WT strain and its isogenic mutant for the iag gene (not for PI-2b) with brain vessels. The authors did not demonstrate a significant difference in the abundance of cell-cell junction proteins (Tjp1, Tjp2, Cldn5, Ocldn, Cdh5, JAM2, JAM3, F11r and Esam), which compose tight junctions and not adherens junctions. This publication differs from our work by (i) analyzing tight junctions and not adherens junctions and, (ii) analyzing the iag gene, known to penetrate endothelial cells in vitro and to play a role in S. agalactiae meningitis in vivo; the authors did not analyze pilus 2b. Our manuscript describes for the first time the influence of PI-2b on the disruption of adherens junctions.

5. Transcriptional Repressor SNAIL-1: The authors should also analyse the possible upregulation of the transcriptional repressor SNAIL-1, as previously reported by Kim et al., J Clin Invest. 2015 leading to the repression of tight junction genes and blood-brain barrier disruption. This would add depth to the interpretation of their findings.

Answer: Endothelial cell junctions are divided into tight, gap, and adherens junctions. The work published by Kim et al., J Clin Invest. 2015, demonstrated the role of SNAIL-1 as a global repressor of tight junction gene expression, inducing increased penetration of S. agalactiae into the BBB. In our work, tight junctions were not analyzed. The focus was on adherens junctions, since VE-cadherin is the main component of endothelial cell-cell adherens junctions that plays a key role in maintaining vascular integrity.

Current, the transcriptional repressor SNAIL-1 is the best-known regulator of cancer invasiveness, physiological and pathological processes including normal embryonic development, repair of epithelial injury, and inducer of epithelial–mesenchymal transition (some publications listed below). In our manuscript, there is no correlation with alterations in the epithelial–mesenchymal transition.

(i) Wieczorek-Szukala K, Lewinski A. The Role of Snail-1 in Thyroid Cancer-What We Know So Far. J Clin Med. 2021 May 26;10(11):2324. doi: 10.3390/jcm10112324;

(ii) Wang L, Li S, Luo H, Lu Q, Yu S. PCSK9 promotes the progression and metastasis of colon cancer cells through regulation of EMT and PI3K/AKT signaling in tumor cells and phenotypic polarization of macrophages. J Exp Clin Cancer Res. 2022 Oct 14;41(1):303. doi: 10.1186/s13046-022-02477-0;

(iii) Chen M, Wu GB, Hua S, Zheng L, Fan Q, Luo M. Dibutyl phthalate (DBP) promotes Epithelial-Mesenchymal Transition (EMT) to aggravate liver fibrosis into cirrhosis and portal hypertension (PHT) via ROS/TGF-β1/Snail-1 signalling pathway in adult rats. Ecotoxicol Environ Saf. 2024 Apr 1;274:116124. doi: 10.1016/j.ecoenv.2024.116124;

(iv) Wang L, Gao J, Zheng S, Luo Z, Xu Z, Che H, Wang Z. CXCR2/Snail-1-Induced Epithelial-Mesenchymal Transition in the Formation and Progression of RCC with Inferior Vena Cava Tumour Thrombus. Arch Esp Urol. 2024 Apr;77(3):292-302. doi: 10.56434/j.arch.esp.urol.20247703.39;

(v) Chiang CH, Yang JD, Liu WL, Chang FY, Yang CJ, Hsu KW, Chiang IT, Hsu FT. Mechanistic insights of lenvatinib: enhancing cisplatin sensitivity, inducing apoptosis, and suppressing metastasis in bladder cancer cells through EGFR/ERK/P38/NF-κB signaling inactivation. Cancer Cell Int. 2025 Feb 15;25(1):47. doi: 10.1186/s12935-024-03597-7.).

6. Figure 2C: Fig. 2C requires clarification. The authors should explain that the positive control (100%) corresponds to FITC dextran translocation in the absence of cells and bacteria. A negative control (0%) should be included, showing the presence of endothelial cells forming tight junctions without bacteria.

Page 10, Lines 216-219: Positive control corresponded to the translocation of FITC-dextran in the absence of HUVECs and bacteria. Negative control corresponded to monolayers of endothelial cells with intact adherens junctions, without bacteria.

7. Figure 3: The relevance of the results shown in Fig. 3 is unclear. If fibrinogen is important for leukocyte transendothelial migration in a VE-cadherin-dependent manner, the authors should explain their conclusions from this experiment and from the previous one showing that GBS disrupt the VE-cadherin. Additionally, they should justify the use of 20 μg of fibrinogen and compare this concentration with 1% BSA.

Pages 12, Lines 270-276: In the presence of fibrinogen, the adhesive capacity of the bacterial strains was increased, especially in HUVECs cultured under shear stress. Furthermore, the GBS90356WT strain showed the highest adhesion in all conditions, especially in the presence of fibrinogen (Fig. 3B; P < 0.001). VE-cadherin staining showed several rupture points and numerous intercell spaces, demonstrating the participation of fibrinogen in the disruption of adherens junctions, especially in HUVECs under shear stress (Fig. 3C; P < 0.001).

Answer: Three concentrations (10 μg, 20 μg and 50 μg of fibrinogen) were tested. The 20 μg concentration showing the best adhesive capacity for the S. agalactiae strains tested.

The concentration of 1% BSA is commonly used as a blocker of additional protein binding sites in ELISA and immunoblotting assays, not interfering with the adhesion of the pathogen to the extracellular matrix or other molecules.

(i) Shin S, Paul-Satyaseela M, Lee JS, Romer LH, Kim KS. Focal adhesion kinase is involved in type III group B streptococcal invasion of human brain microvascular endothelial cells. Microb Pathog. 2006 Oct-Nov;41(4-5):168-73. doi: 10.1016/j.micpath.2006.07.003;

(ii) Tancioni I, Miller NL, Uryu S, Lawson C, Jean C, Chen XL, Kleinschmidt EG, Schlaepfer DD. FAK activity protects nucleostemin in facilitating breast cancer spheroid and tumor growth. Breast Cancer Res. 2015 Mar 28;17:47. doi: 10.1186/s13058-015-0551-x;

(iii) Matyas GR, Rao M, Alving CR. Induction and detection of antibodies to squalene. II. Optimization of the assay for murine antibodies. J Immunol Methods. 2002 Sep 15;267(2):119-29. doi: 10.1016/s0022-1759(02)00180-1.

## REVIEWER #3

Overall, the manuscript is well-written and the content contributes to the knowledge of GBS pathogenesis. I have no special concern of any section of the manuscript. Below, a list of recommendations and suggestions.

Page 4

L 16: "stillbirths deaths" is redundant. Are authors supposed to mention "stillbirths and infant deaths"? In addition, it would be valuable to include a reference for these epidemiological data.

Answer: "stillbirths deaths" was changed and the reference was included.

Page 4, Lines 74-75: S. agalactiae causes at least 400,000 maternal and neonatal infections, accounting for approximately 50,000 stillbirths and 50,000 to 100,000 infant deaths annually (Gonçalves et al., 2022).

L 25-27: "virulence factors that can disrupt the endothelial barrier, which can directly kill endothelial cells and/or disrupt endothelial cell-cell junctions during sepsis"… it is somewhat repetitive. I suggest improving this sentence.

Answer: The sentence has been improved.

Page 4, Lines 79-82: Many bacteria produce virulence factors that can disrupt the endothelial barrier through a variety of mechanisms, including direct damage of endothelial cells, alteration of the endothelial cytoskeleton, and disruption of cell-cell junctions during sepsis.

L57: "morphology, cell signaling,…" the word "AND" is needed after "signaling".

Answer: The word "AND" was included after "signaling".

Page 5, Line 99: … the expression of surface molecules, morphology, cell signaling, and interactions with different pathogens…

Page 8

L41: Please correct: MILLICELL

Answer: The word "MILLICELL" was corrected.

Page 11

L30-32: "characteristic of endothelial cell alignment that is critical for vascular homeostasis..." The sentence is too large and can be shortened. I suggest removing "that is".

Page 12, Lines 282-284: Endothelial cells are exposed to shear stress, a fundamental physical characteristic of endothelial cell alignment, critical for vascular homeostasis.

L45: "severe invasive infant disease..." I suggest changing to "neonatal disease".

Page 13, Line 290: … neonatal disease…

L53: "advantage in the human host by reducing host immune responses…" The sentence is too large and can be shortened. I suggest that the authors revise and improve it.

Page 13, Lines 291-294: PI-2b expression is regulated in S. agalactiae ST-17 strains by a 43-bp hairpin-like structure in the upstream region of PI-2b operon, conferring a selective advantage in the human host, either by reducing host immune responses or by increasing its dissemination potential.

L57: "but may also be present in non-ST17 human strains..." it is somewhat strange "human strains". I suggest that the authors revise and improve it.

Page 13, Lines 294-295: …but may also be present in non-ST17 strains isolated from humans (Springman et al., . 2014).

Page 12

L37: I suggest removing "Epidemiological." Despite the large sample from different countries, the study has focused on investigating antibodies against GBS cps and pilus.

Answer: "Epidemiological" has been removed from the text.

Page 14, Lines 309-311: S. agalactiae strains isolated from neonatal invasive infections in European countries showed high expression of PI-2b compared with PI-1.

Page 13

L25-27: "Decreased expression of VE-cadherin by the GBS90356Δpilus2b strain was less than the reduction caused by the GBS90356WT strain, demonstrating the involvement of PI-2b". I suggest the authors improve this sentence, especially the end, "involvement of PI-2b".

Page 14, Lines 327-330: The decrease in VE-cadherin expression by the GBS90356Δpilus2b strain was lower when compared to the GBS90356WT strain, demonstrating the role of PI-2b during disease progression.

L29-30: I suggest the authors improve the sentence. They should initiate this paragraph by clarifying that the text is a result of other studies, since "The authors" appears in the middle of the paragraph, but the reader has no idea until he reaches this point. Another alternative is simply remove "The authors", reorganize the sentence, and reference it.

Pages 14-15, Lines 331-334: A previous study demonstrated that adherens junction VE-cadherin controls tight junction organization by promoting increased permeability. The mechanism occurs through upregulation of tight junction claudin-5, mediated by downregulation of VE-cadherin (Taddei et al., 2008).

L41: "explaining" should be changed to "explains"

Page 15, Lines 335-337: The link between adherens junction and tight junctions explains why VE-cadherin inhibition can cause a marked increase in permeability.

L37-39: "Previous data demonstrated that fibrinogen resulted in disruption of endothelial barrier integrity…" I recommend the authors to improve the sentence: the original reference mentions fibrinogen fragments, degradation products of fibrinogen, not fibrinogen itself. So, the subsequent discussion should be carefully revised.

Page 16, Lines 366-373: Vascular endothelial barrier disruption is a central mechanism in the development of multiple organ failure after severe trauma and hemorrhage. Fibrinolysis occurs during organ dysfunction, resulting in high circulating levels of fibrinogen degradation products, such as fragment X that induces endothelial cell hyperpermeability by reducing VE-cadherin expression. In addition to affecting VE-cadherin, fragment X also induced significant changes in genes that regulate endothelial cell-mediated coagulation, inflammation, angiogenesis, and vasoconstriction (Olson et al., 2024). Thus, fibrinogen and/or its fragments increased the severity of several inflammatory conditions.

EDITOR COMMENTS

In addition to responding to all points raised by reviewers, it is necessary to:

(i) Improve the methodology sections with more details of some experiments performed as suggested by the reviewers. In addition, in the section of RT-PCR: "cDNA synthesis and PCR were performed as described previously." There is no reference, and the authors need to briefly describe quantities and conditions used in the reactions.

Page 9, Lines 191-207: HUVECs were collected by a cell-scraper after S. agalactiae infection, and total RNA was isolated using TRIzol reagent (Invitrogen, São Paulo, SP, Brazil) according to the manufacturer's protocol. The purity (A260/A280) and concentration of RNA were determined using a NanoDrop 2000 spectrophotometer (Thermo Fisher Scientific, São Paulo, SP, Brazil). Using the Maxima First Strand cDNA Synthesis Kit with dsDNase (Thermo Fisher Scientific), reverse transcription of cDNA was carried out in accordance with the manufacturer's instructions. Real-time reverse transcription PCR (RT-PCR) amplification was performed with SYBR Green Master Mix in a StepOne PCR amplifier (Thermo Fisher Scientific). Untreated HUVECs were used as the reference sample and β-actin was used as the endogenous control. The PCR primer sequences and annealing temperatures for PCR were: VE-cadherin (230 bp, Tm 55°C and 30 cycles) Forward 5'ACATCACAGTCAAGTAT-GGGC3' Reverse 5'GATGCAGAGTAAGATGGCTGC 3' and β-actin (289 bp, Tm 58°C and 25 cycles) Forward 5' TGGACTTCGAGCAAGAGATGG3' Reverse 5'ATCTCTTCTGCATCCTGTCG3' as described before (Yu et al., 2017). The amplified DNA products were separated on 2% agarose gel, stained with ethidium bromide, visualized and photographed. The assay was performed in triplicate, and each experiment was repeated at least three times.

(ii) The discussion should be carefully revised with the reviewers' suggestions, including more articles from the literature.

Answer: The discussion was revised with the reviewers' suggestions, including more articles from the literature.

(iii) Page 15: "In this work, the presence of fbsA, fbsB, Srr1 and Srr2 genes were detected in GBS90356 strain (data not shown)." It would be important to include this data as a Supplementary Figure of the manuscript.

Answer: These results are part of a larger study of virulence factor and antimicrobial resistance genes from the GBS90356 strain that have been submitted for publication and are awaiting reviewers' comments. I cannot include these data as supplementary.

## SECOND REVIEW ROUND

### REVIEWERS' COMMENTS

#### REVIEWER #1

Reviewer comments: The authors have satisfied this reviewers comments and the adjustments made in response to all reviewers has strengthened the manuscript.

#### REVIEWER #2

Reviewer comments: no comments.

