## [Reviewer Report · FIRST REVIEW ROUND - REVIEWERS COMMENTS]

## REVIEWER #1

a - The abstract is informative;

b - The work presented in this manuscript is a good contribution to the streptococcus agalactiae field. It provides further mechanisms for the field to understand the ability of the organism to adhere and disrupt the endothelial barrier;

c - Methodology:

- It would be useful for the authors to provide more details regarding the sheer stress experimental set up. Explaining what manufacturers system was used to maintain the constant flow, how long the cells were under the sheer stress before imaging, how were the cells infected in the system, etc.

- In the statistic section it is stated that the figures show mean and SEM but all the figure legends state mean and SD. Can the authors clarify this.

d - References are fine;

e - Figures:

- In Figure 2 it would be beneficial to increase the “a-f” lettering in the microscope images as these are very difficult to see currently.

- In each figure legend there is a typo for the GBS WT strain (9056WT instead of 90356WT).

## REVIEWER #2

In their study, the authors demonstrate that the hypervirulent Group B Streptococcus strain GBS90356 exhibits enhanced adherence (a 4-5 fold increase) to human vein endothelial cells (HUVEC) under shear stress conditions. This adherence is reduced when using a PI-2b pilus mutant. The findings also suggest that Group B Streptococcus can disrupt cell junctions by decreasing VE-cadherin expression under shear stress conditions and that the addition of external fibrinogen further enhances GBS adherence to endothelial cells. However, the overall rationale of the manuscript could be clearer, and the experimental model using HUVEC, as well as the results presented in Figures 2 and 3, may benefit from additional validation and explanation.

The title of the manuscript is misleading and not backed up by the results.

Main Comments:

1. Strain Selection: The authors focus on a single GBS strain, GBS90356, which was isolated from a meningitis case and is not well-characterized. To strengthen their findings, it would be beneficial to include additional strains, such as another CC-17 strain like COH1 and its isogenic COH1∆PI2b mutant, for comparison.

2. Rationale for PI2b Mutant: The choice to test the PI2b mutant should be better explained. Previous studies by Lazzarin et al. and Perichon et al. have shown the role of PI2b in adhesion to epithelial alveolar cells (A549) and human brain microvascular endothelial cells (hBMEC). These findings could provide a stronger context for the current study.

3. Fibrinogen Adherence: It is well-established that CC-17 strains adhere better to fibrinogen, primarily due to the surface expression of the Srr2 glycoprotein (Six et al., Molecular Microbiology 2015). Including this reference in the discussion would enrich the manuscript.

4. VE-Cadherin Downregulation: The immunofluorescence images showing VE-cadherin downregulation upon GBS infection could be more convincing. Including other proteins that constitute cell junctions for analysis would provide a more comprehensive view. Additionally, a study by Campeau et al. (2020) performed proteome profiling of blood-brain barrier perturbation by GBS and did not report major alterations of endothelial junctions in vivo. This study should be considered in the discussion.

5. Transcriptional Repressor SNAIL-1: The authors should also analyse the possible upregulation of the transcriptional repressor SNAIL-1, as previously reported by Kim et al., J Clin Invest. 2015 leading to the repression of tight junction genes and blood-brain barrier disruption. This would add depth to the interpretation of their findings.

6. Figure 2C: Fig. 2C requires clarification. The authors should explain that the positive control (100%) corresponds to FITC dextran translocation in the absence of cells and bacteria. A negative control (0%) should be included, showing the presence of endothelial cells forming tight junctions without bacteria. The effect of adding bacteria to the filter should also be demonstrated.

7. Figure 3: The relevance of the results shown in Fig. 3 is unclear. If fibrinogen is important for leukocyte transendothelial migration in a VE-cadherin-dependent manner, the authors should explain their conclusions from this experiment and from the previous one showing that GBS disrupt the VE-cadherin. Additionally, they should justify the use of 20 µg of fibrinogen and compare this concentration with $1\%$ BSA.

## REVIEWER #3

Overall, the manuscript is well-written and the content contributes to the knowledge of GBS pathogenesis. I have no special concern of any section of the manuscript. Below, a list of recommendations and suggestions.

Page 4:

L 16: “stillbirths deaths” is redundant. Are authors supposed to mention “stillbirths and infant deaths”? In addition, it would be valuable to include a reference for these epidemiological data.

L 25-27: “virulence factors that can disrupt the endothelial barrier, which can directly kill endothelial cells and/or disrupt endothelial cell-cell junctions during sepsis”… it is somewhat repetitive. I suggest improving this sentence.

L57: “morphology, cell signaling,…” the word “AND” is needed after “signaling”.

Page 8:

L41: Please correct: MILLICELL

Page 11:

L30-32: “characteristic of endothelial cell alignment that is critical for vascular homeostasis...” The sentence is too large and can be shortened. I suggest removing “that is”.

L45: “severe invasive infant disease...” I suggest changing to “neonatal disease”.

L53: “advantage in the human host by reducing host immune responses…” The sentence is too large and can be shortened. I suggest that the authors revise and improve it.

L57: “but may also be present in non-ST17 human strains...” it is somewhat strange “human strains”. I suggest that the authors revise and improve it.

Page 12:

L37: I suggest removing “Epidemiological.” Despite the large sample from different countries, the study has focused on investigating antibodies against GBS cps and pilus.

Page 13:

L25-27: “Decreased expression of VE-cadherin by the GBS90356$\Delta$pilus2b strain was less than the reduction caused by the GBS90356WT strain, demonstrating the involvement of PI-2b”. I suggest the authors improve this sentence, especially the end, “involvement of PI-2b”.

L29-30: I suggest the authors improve the sentence. They should initiate this paragraph by clarifying that the text is a result of other studies, since “The authors” appears in the middle of the paragraph, but the reader has no idea until he reaches this point. Another alternative is simply remove “The authors”, reorganize the sentence, and reference it.

L41: “explaining” should be changed to “explains”.

Page 14:

L37-39: “Previous data demonstrated that fibrinogen resulted in disruption of endothelial barrier integrity…” I recommend the authors to improve the sentence: the original reference mentions fibrinogen fragments, degradation products of fibrinogen, not fibrinogen itself. So, the subsequent discussion should be carefully revised.

## EDITOR COMMENTS

In addition to responding to all points raised by reviewers, it is necessary to:

(i) improve the methodology sections with more details of some experiments performed as suggested by the reviewers. In addition, in the section of RT-PCR: “cDNA synthesis and PCR were performed as described previously.” There is no reference, and the authors need to briefly describe quantities and conditions used in the reactions.

(ii) The discussion should be carefully revised with the reviewers suggestions, including more articles from the literature.

(iii) Page 15: “In this work, the presence of fbsA, fbsB, Srr1 and Srr2 genes were detected in GBS90356 strain (data not shown).” It would be important to include this data as a Supplementary Figure of the manuscript.

---

## [Author Response · AUTHORS' RESPONSE TO REVIEWERS]

Dr. Geovane Dias-Lopes

Handling Editor

Memórias do Instituto Oswaldo Cruz

Dear Dr. Dias-Lopes,

Thank you for your message and critical comments concerning the manuscript ID MIOC-2025-0077 entitled “Streptococcus agalactiae disrupts VE-cadherin intercellular junctions and induces endothelial barrier dysfunction”” by Oliveira and co-workers. Following the suggestions made in the first version submitted to Memórias do Instituto Oswaldo Cruz, we are submitting the revised version. The manuscript was revised and supplied with new information to improve data quality, as pointed out by the reviewers.

Sincerely,

Professor Dr Prescilla Emy Nagao

---

## [Reviewer Report · FIRST REVIEW ROUND]

## REVIEWER #1

a - The abstract is informative.

b - The work presented in this manuscript is a good contribution to the Streptococcus agalactiae field. It provides further mechanisms for the field to understand the ability of the organism to adhere and disrupt the endothelial barrier.

c - Methodology:

- It would be useful for the authors to provide more details regarding the sheer stress experimental set up. Explaining what manufacturers system was used to maintain the constant flow, how long the cells were under the sheer stress before imaging, how were the cells infected in the system, etc.

Page 7, lines 143-152: HUVECs were submitted to steady laminar shear stress as previously described (Colgan et al., 2007). Briefly, HUVECs were seeded at $8\times 10^5$ cells/well in regular six-well plates and allowed to reach confluence, typically in 2-3 days. HUVECs were exposed to $10\text{ dyn/cm}^2$ of constant shear stress for 24 h in orbital rotator (CO2 Resistant Shaker, Thermo Scientific™) using the equation $\tau_w = \alpha\sqrt{rh(2\pi f)^3}$, where $\tau_w$ is the shear stress, $\alpha$ is the radius of rotation (cm), $r$ is the density of the liquid (g/L), $h$ is ﬂuid viscosity ($0.0075\text{ dyn/cm}^2$ at $37^\circ\text{C}$), and $f$ is the rotation per second. The flow device was kept at $37^\circ\text{C}$ for 24 h in a $5\%$ $\text{CO}_2$ atmosphere. Each experimental condition was repeated three times. After shear stress, HUVECs monolayers were infected with S. agalactiae without orbital rotation as described below.

- In the statistic section it is stated that the figures show mean and SEM but all the figure legends state mean and SD. Can the authors clarify this.

Answer: There was a typing error. The results were expressed as “Mean and Standard Error of the Mean (SEM) and not Standard Deviation (SD). The figure legends have been corrected.

d - References are fine.

e - Figures:

- In Figure 2 it would be beneficial to increase the “a-f” lettering in the microscope images as these are very difficult to see currently.

- In each figure legend there is a typo for the GBS WT strain (9056WT instead of 90356WT).

Answer: Figure 2 has been corrected.

## REVIEWER #2

In their study, the authors demonstrate that the hypervirulent Group B Streptococcus strain GBS90356 exhibits enhanced adherence (a 4-5 fold increase) to human vein endothelial cells (HUVEC) under shear stress conditions. This adherence is reduced when using a PI-2b pilus mutant. The findings also suggest that Group B Streptococcus can disrupt cell junctions by decreasing VE-cadherin expression under shear stress conditions and that the addition of external fibrinogen further enhances GBS adherence to endothelial cells. However, the overall rationale of the manuscript could be clearer, and the experimental model using HUVEC, as well as the results presented in Figures 2 and 3, may benefit from additional validation and explanation.

The title of the manuscript is misleading and not backed up by the results.

Answer: Title was changed to “Infection of endothelial cells by Streptococcus agalactiae reveals potential role of PI-2b pilus on endothelial barrier dysfunction”.

Main Comments:

1. Strain Selection: The authors focus on a single GBS strain, GBS90356, which was isolated from a meningitis case and is not well-characterized. To strengthen their findings, it would be beneficial to include additional strains, such as another CC-17 strain like COH1 and its isogenic COH1∆PI2b mutant, for comparison.

Answer: Strain GBS90356 is well characterized and has its whole genome sequenced and published by our group as the first Brazilian ST-17 strain sequenced (Lannes-Costa PS, Baraúna RA, Ramos JN, Veras JFC, Conceição MVR, Vieira VV, de Mattos-Guaraldi AL, Ramos RTJ, Doran KS, Silva A, Nagao PE. Comparative genomic analysis and identification of pathogenicity islands of hypervirulent ST-17 Streptococcus agalactiae Brazilian strain. Infect Genet Evol. 2020 Jun;80:104195. doi: 10.1016/j.meegid.2020.104195). GBS90356 strain was also studied for virulence properties (Costa et al. 2011; Oliveira et al., 2018; Lannes-Costa et al., 2020, da Conceição et al., 2023; Lannes-Costa et al., 2024). The reviewer's suggestion to include COH1 strain and its isogenic mutant COH1∆PI2b for comparison is not relevant in this experimental assay. Published studies evaluating the importance of pilus 2b (or other molecules such as SaeRS, LytSR, BvaP, SfpB) in the virulence of S. agalactiae did not use more than one wild-type strain and its isogenic mutant belonging to ST-17 or another capsular type of a clinical isolate of S. agalactiae (some publications listed below). This is currently justified because the comparisons involve other factors and variables specific to each strain belonging to the same clone (already demonstrated by proteomics and metabolomics), which can significantly interfere in the final analysis of the results obtained. Our work highlights the importance of analyzing the virulence profile of the first Brazilian ST-17 strain sequenced in Brazil (GBS90356), whose data demonstrate for the first time the pathogenic potential of dissemination and disruption of adherens junctions by S. agalactiae, mainly in endothelial cells under physiological shear stress.

(i) Lazzarin et al., 2017. Contribution of pilus type 2b to invasive disease caused by a Streptococcus agalactiae ST-17 strain. BMC Microbiol. 3;17(1):148. doi: 10.1186/s12866-017-1057-8;

(ii) Campeau A, Mills RH, Blanchette M, Bajc K, Malfavon M, Munji RN, Deng L, Hancock B, Patras KA, Olson J, Nizet V, Daneman R, Doran K, Gonzalez DJ. Multidimensional Proteome Profiling of Blood-Brain Barrier Perturbation by Group B Streptococcus. mSystems. 2020 Aug 25;5(4):e00368-20. doi: 10.1128/mSystems.00368-20;

(iii) Thomas LS, Cook LC. A Novel Conserved Protein in Streptococcus agalactiae, BvaP, Is Important for Vaginal Colonization and Biofilm Formation. mSphere. 2022 Dec 21;7(6):e0042122. doi: 10.1128/msphere.00421-22;

(iv) Li S, Li W, Liang Q, Cao J, Li H, Li Z, Li A. Characterization and virulence of Streptococcus agalactiae deficient in SaeRS of the two-component system. Front Microbiol. 2023 Apr 17;14:1121621. doi: 10.3389/fmicb.2023.1121621;

(v) AlQadeeb H, Baltazar M, Cazares A, Poonpanichakul T, Kjos M, French N, Kadioglu A, O'Brien M. The Streptococcus agalactiae LytSR two-component regulatory system promotes vaginal colonization and virulence in vivo. Microbiol Spectr. 2024 Nov 5;12(11):e0197024. doi: 10.1128/spectrum.01970-24;

(vi) Li H, Cao J, Han Q, Li Z, Zhuang J, Wang C, Wang H, Luo Z, Wang B, Li A. Protease SfpB plays an important role in cell membrane stability and immune system evasion in Streptococcus agalactiae. Microb Pathog. 2024 Jul;192:106683. doi: 10.1016/j.micpath.2024.106683.

Page 6, Lines 124-125: GBS90356 strain was the first Brazilian ST-17 strain sequenced in Brazil by our group and was partially investigated for virulence properties (Costa et al. 2011; Oliveira et al., 2018; Lannes-Costa et al., 2020, da Conceição et al., 2023; Lannes-Costa et al., 2024).

2. Rationale for PI2b Mutant: The choice to test the PI2b mutant should be better explained. Previous studies by Lazzarin et al. and Perichon et al. have shown the role of PI2b in adhesion to epithelial alveolar cells (A549) and human brain microvascular endothelial cells (hBMEC). These findings could provide a stronger context for the current study.

Page 4, Lines 83-90: In S. agalactiae, pilus type 2b (PI-2b) mediates adhesion and invasion of brain endothelial cells and contributes to translocation across the blood-brain barrier (Lazzarin et al., 2017). PI-2b also contributes to the pathogenesis of S. agalactiae infection by mediating invasion in several human epithelial cell lines (pulmonary A549, cervical HeLa, and colonic C2BBe1) (Adderson et al., 2003; Périchon et al., 2019). In addition, the presence of PI-2b increased phagocytosis of S. agalactiae by murine and human macrophages (Chattopadhyay et al. 2011). However, the pathway by which S. agalactiae crosses the endothelial barrier remains unclear.

3. Fibrinogen Adherence: It is well-established that CC-17 strains adhere better to fibrinogen, primarily due to the surface expression of the Srr2 glycoprotein (Six et al., Molecular Microbiology 2015). Including this reference in the discussion would enrich the manuscript.

Answer: The reference Six et al., Molecular Microbiology, 2015 was included in the manuscript discussion.

Page 15, Lines 346-353: Consistently, S. agalactiae ST-17 clinical isolates exhibit a higher capacity to bind fibrinogen when compared with non-ST-17 strains (Dramsi et al., 2012). Srr2 has a higher affinity for fibrinogen than Srr1, suggesting that its expression may be critical for the hypervirulence of ST-17 isolates. Adhesion to fibrinogen is essential for early colonization of host tissues and organs. Invading bacterial pathogens frequently bind to plasminogen/plasmin, and the resulting increase in cell surface proteolytic activity contributes to their subsequent dissemination within the infected host by facilitating the crossing of barriers such as the blood-brain barrier (Six et al., 2015).

4. VE-Cadherin Downregulation: The immunofluorescence images showing VE-cadherin downregulation upon GBS infection could be more convincing. Including other proteins that constitute cell junctions for analysis would provide a more comprehensive view. Additionally, a study by Campeau et al. (2020) performed proteome profiling of blood-brain barrier perturbation by GBS and did not report major alterations of endothelial junctions in vivo. This study should be considered in the discussion.

Answer: Campeau et al. (2020) compared the COH1WT strain and its isogenic mutant for the *iag* gene (not for PI-2b) with brain vessels. The authors did not demonstrate a significant difference in the abundance of cell-cell junction proteins ($\text{Tjp}1$, $\text{Tjp}2$, $\text{Cldn}5$, $\text{Ocldn}$, $\text{Cdh}5$, $\text{JAM}2$, $\text{JAM}3$, $\text{F}11\text{r}$ and $\text{Esam}$), which compose tight junctions and not adherens junctions. This publication differs from our work by (i) analyzing tight junctions and not adherens junctions and, (ii) analyzing the *iag* gene, known to penetrate endothelial cells in vitro and to play a role in S. agalactiae meningitis in vivo; the authors did not analyze pilus 2b. Our manuscript describes for the first time the influence of PI-2b on the disruption of adherens junctions.

5. Transcriptional Repressor SNAIL-1: The authors should also analyse the possible upregulation of the transcriptional repressor SNAIL-1, as previously reported by Kim et al., J Clin Invest. 2015 leading to the repression of tight junction genes and blood-brain barrier disruption. This would add depth to the interpretation of their findings.

Answer: Endothelial cell junctions are divided into tight, gap, and adherens junctions. The work published by Kim et al., J Clin Invest. 2015, demonstrated the role of SNAIL-1 as a global repressor of tight junction gene expression, inducing increased penetration of S. agalactiae into the BBB. In our work, tight junctions were not analyzed. The focus was on adherens junctions, since VE-cadherin is the main component of endothelial cell-cell adherens junctions that plays a key role in maintaining vascular integrity.

Current, the transcriptional repressor SNAIL-1 is the best-known regulator of cancer invasiveness, physiological and pathological processes including normal embryonic development, repair of epithelial injury, and inducer of epithelial-mesenchymal transition (some publications listed below). In our manuscript, there is no correlation with alterations in the epithelial-mesenchymal transition.

(i) Wieczorek-Szukala K, Lewinski A. The Role of Snail-1 in Thyroid Cancer-What We Know So Far. J Clin Med. 2021 May 26;10(11):2324. doi: 10.3390/jcm10112324;

(ii) Wang L, Li S, Luo H, Lu Q, Yu S. PCSK9 promotes the progression and metastasis of colon cancer cells through regulation of EMT and PI3K/AKT signaling in tumor cells and phenotypic polarization of macrophages. J Exp Clin Cancer Res. 2022 Oct 14;41(1):303. doi: 10.1186/s13046-022-02477-0;

(iii) Chen M, Wu GB, Hua S, Zheng L, Fan Q, Luo M. Dibutyl phthalate (DBP) promotes Epithelial-Mesenchymal Transition (EMT) to aggravate liver fibrosis into cirrhosis and portal hypertension (PHT) via ROS/TGF-β1/Snail-1 signalling pathway in adult rats. Ecotoxicol Environ Saf. 2024 Apr 1;274:116124. doi: 10.1016/j.ecoenv.2024.116124;

(iv) Wang L, Gao J, Zheng S, Luo Z, Xu Z, Che H, Wang Z. CXCR2/Snail-1-Induced Epithelial-Mesenchymal Transition in the Formation and Progression of RCC with Inferior Vena Cava Tumour Thrombus. Arch Esp Urol. 2024 Apr;77(3):292-302. doi: 10.56434/j.arch.esp.urol.20247703.39;

(v) Chiang CH, Yang JD, Liu WL, Chang FY, Yang CJ, Hsu KW, Chiang IT, Hsu FT. Mechanistic insights of lenvatinib: enhancing cisplatin sensitivity, inducing apoptosis, and suppressing metastasis in bladder cancer cells through EGFR/ERK/P38/NF-κB signaling inactivation. Cancer Cell Int. 2025 Feb 15;25(1):47. doi: 10.1186/s12935-024-03597-7.

6. Figure 2C: Fig. 2C requires clarification. The authors should explain that the positive control ($100\%$) corresponds to FITC dextran translocation in the absence of cells and bacteria. A negative control ($0\%$) should be included, showing the presence of endothelial cells forming tight junctions without bacteria.

Page 10, Lines 216-219: Positive control corresponded to the translocation of FITC-dextran in the absence of HUVECs and bacteria. Negative control corresponded to monolayers of endothelial cells with intact adherens junctions, without bacteria.

7. Figure 3: The relevance of the results shown in Fig. 3 is unclear. If fibrinogen is important for leukocyte transendothelial migration in a VE-cadherin-dependent manner, the authors should explain their conclusions from this experiment and from the previous one showing that GBS disrupt the VE-cadherin. Additionally, they should justify the use of 20 µg of fibrinogen and compare this concentration with $1\%$ BSA.

Pages 12, Lines 270-276: In the presence of fibrinogen, the adhesive capacity of the bacterial strains was increased, especially in HUVECs cultured under shear stress. Furthermore, the $\text{GBS}90356\text{WT}$ strain showed the highest adhesion in all conditions, especially in the presence of fibrinogen (Fig. 3B; P < 0.001). VE-cadherin staining showed several rupture points and numerous intercell spaces, demonstrating the participation of fibrinogen in the disruption of adherens junctions, especially in HUVECs under shear stress (Fig. 3C; P < 0.001).

Answer: Three concentrations (10 µg, 20 µg and 50 µg of fibrinogen) were tested. The 20 µg concentration showing the best adhesive capacity for the S. agalactiae strains tested.

The concentration of $1\%$ BSA is commonly used as a blocker of additional protein binding sites in ELISA and immunoblotting assays, not interfering with the adhesion of the pathogen to the extracellular matrix or other molecules.

(i) Shin S, Paul-Satyaseela M, Lee JS, Romer LH, Kim KS. Focal adhesion kinase is involved in type III group B streptococcal invasion of human brain microvascular endothelial cells. Microb Pathog. 2006 Oct-Nov;41(4-5):168-73. doi: 10.1016/j.micpath.2006.07.003;

(ii) Tancioni I, Miller NL, Uryu S, Lawson C, Jean C, Chen XL, Kleinschmidt EG, Schlaepfer DD. FAK activity protects nucleostemin in facilitating breast cancer spheroid and tumor growth. Breast Cancer Res. 2015 Mar 28;17:47. doi: 10.1186/s13058-015-0551-x;

(iii) Matyas GR, Rao M, Alving CR. Induction and detection of antibodies to squalene. II. Optimization of the assay for murine antibodies. J Immunol Methods. 2002 Sep 15;267(2):119-29. doi: 10.1016/s0022-1759(02)00180-1.

## REVIEWER #3

Overall, the manuscript is well-written and the content contributes to the knowledge of GBS pathogenesis. I have no special concern of any section of the manuscript. Below, a list of recommendations and suggestions.

Page 4

L 16: “stillbirths deaths” is redundant. Are authors supposed to mention “stillbirths and infant deaths”? In addition, it would be valuable to include a reference for these epidemiological data.

Answer: “stillbirths deaths” was changed and the reference was included.

Page 4, Lines 74-75: S. agalactiae causes at least 400,000 maternal and neonatal infections, accounting for approximately 50,000 stillbirths and 50,000 to 100,000 infant deaths annually (Gonçalves et al., 2022).

L 25-27: “virulence factors that can disrupt the endothelial barrier, which can directly kill endothelial cells and/or disrupt endothelial cell-cell junctions during sepsis”… it is somewhat repetitive. I suggest improving this sentence.

Answer: The sentence has been improved.

Page 4, Lines 79-82: Many bacteria produce virulence factors that can disrupt the endothelial barrier through a variety of mechanisms, including direct damage of endothelial cells, alteration of the endothelial cytoskeleton, and disruption of cell-cell junctions during sepsis.

L57: “morphology, cell signaling,…” the word “AND” is needed after “signaling”.

Answer: The word “AND” was included after “signaling”.

Page 5, Line 99: … the expression of surface molecules, morphology, cell signaling, and interactions with different pathogens…

Page 8

L41: Please correct: MILLICELL

Answer: The word “MILLICELL” was corrected.

Page 11

L30-32: “characteristic of endothelial cell alignment that is critical for vascular homeostasis...” The sentence is too large and can be shortened. I suggest removing “that is”.

Page 12, Lines 282-284: Endothelial cells are exposed to shear stress, a fundamental physical characteristic of endothelial cell alignment, critical for vascular homeostasis.

L45: “severe invasive infant disease...” I suggest changing to “neonatal disease”.

Page 13, Line 290: … neonatal disease…

L53: “advantage in the human host by reducing host immune responses…” The sentence is too large and can be shortened. I suggest that the authors revise and improve it.

Page 13, Lines 291-294: $\text{PI-2b}$ expression is regulated in S. agalactiae ST-17 strains by a 43-bp hairpin-like structure in the upstream region of $\text{PI-2b}$ operon, conferring a selective advantage in the human host, either by reducing host immune responses or by increasing its dissemination potential.

L57: “but may also be present in non-ST17 human strains...” it is somewhat strange “human strains”. I suggest that the authors revise and improve it.

Page 13, Lines 294-295: …but may also be present in non-ST17 strains isolated from humans (Springman et al., . 2014).

Page 12

L37: I suggest removing “Epidemiological.” Despite the large sample from different countries, the study has focused on investigating antibodies against GBS cps and pilus.

Answer: “Epidemiological” has been removed from the text.

Page 14, Lines 309-311: S. agalactiae strains isolated from neonatal invasive infections in European countries showed high expression of $\text{PI-2b}$ compared with $\text{PI-1}$.

Page 13

L25-27: “Decreased expression of VE-cadherin by the $\text{GBS}90356\Delta\text{pilus}2\text{b}$ strain was less than the reduction caused by the $\text{GBS}90356\text{WT}$ strain, demonstrating the involvement of PI-2b”. I suggest the authors improve this sentence, especially the end, “involvement of PI-2b”.

Page 14, Lines 327-330: The decrease in VE-cadherin expression by the $\text{GBS}90356\Delta\text{pilus}2\text{b}$ strain was lower when compared to the $\text{GBS}90356\text{WT}$ strain, demonstrating the role of $\text{PI-2b}$ during disease progression.

L29-30: I suggest the authors improve the sentence. They should initiate this paragraph by clarifying that the text is a result of other studies, since “The authors” appears in the middle of the paragraph, but the reader has no idea until he reaches this point. Another alternative is simply remove “The authors”, reorganize the sentence, and reference it.

Pages 14-15, Lines 331-334: A previous study demonstrated that adherens junction VE-cadherin controls tight junction organization by promoting increased permeability. The mechanism occurs through upregulation of tight junction claudin-5, mediated by downregulation of VE-cadherin (Taddei et al., 2008).

L41: “explaining” should be changed to “explains”

Page 15, Lines 335-337: The link between adherens junction and tight junctions explains why VE-cadherin inhibition can cause a marked increase in permeability.

L37-39: “Previous data demonstrated that fibrinogen resulted in disruption of endothelial barrier integrity…” I recommend the authors to improve the sentence: the original reference mentions fibrinogen fragments, degradation products of fibrinogen, not fibrinogen itself. So, the subsequent discussion should be carefully revised.

Page 16, Lines 366-373: Vascular endothelial barrier disruption is a central mechanism in the development of multiple organ failure after severe trauma and hemorrhage. Fibrinolysis occurs during organ dysfunction, resulting in high circulating levels of fibrinogen degradation products, such as fragment X that induces endothelial cell hyperpermeability by reducing VE-cadherin expression. In addition to affecting VE-cadherin, fragment X also induced significant changes in genes that regulate endothelial cell-mediated coagulation, inflammation, angiogenesis, and vasoconstriction (Olson et al., 2024). Thus, fibrinogen and/or its fragments increased the severity of several inflammatory conditions.

## EDITOR COMMENTS

In addition to responding to all points raised by reviewers, it is necessary to:

(i) Improve the methodology sections with more details of some experiments performed as suggested by the reviewers. In addition, in the section of RT-PCR: “cDNA synthesis and PCR were performed as described previously.” There is no reference, and the authors need to briefly describe quantities and conditions used in the reactions.

Page 9, Lines 191-207: HUVECs were collected by a cell-scraper after S. agalactiae infection, and total RNA was isolated using TRIzol reagent (Invitrogen, São Paulo, SP, Brazil) according to the manufacturer's protocol. The purity ($\text{A}_{260}/\text{A}_{280}$) and concentration of RNA were determined using a NanoDrop 2000 spectrophotometer (Thermo Fisher Scientific, São Paulo, SP, Brazil). Using the Maxima First Strand cDNA Synthesis Kit with dsDNase (Thermo Fisher Scientific), reverse transcription of cDNA was carried out in accordance with the manufacturer's instructions. Real-time reverse transcription PCR (RT-PCR) amplification was performed with SYBR Green Master Mix in a StepOne PCR amplifier (Thermo Fisher Scientific). Untreated HUVECs were used as the reference sample and $\beta$-actin was used as the endogenous control. The PCR primer sequences and annealing temperatures for PCR were: VE-cadherin (230 bp, $\text{T}_{\text{m}}$ $55^\circ\text{C}$ and 30 cycles) Forward 5'ACATCACAGTCAAGTATGGGC3' Reverse 5'GATGCAGAGTAAGATGGCTGC 3' and $\beta$-actin (289 bp, $\text{T}_{\text{m}}$ $58^\circ\text{C}$ and 25 cycles) Forward 5' TGGACTTCGAGCAAGAGATGG3' Reverse 5'ATCTCTTCTGCATCCTGTCG3' as described before (Yu et al., 2017). The amplified DNA products were separated on $2\%$ agarose gel, stained with ethidium bromide, visualized and photographed. The assay was performed in triplicate, and each experiment was repeated at least three times.

(ii) The discussion should be carefully revised with the reviewers suggestions, including more articles from the literature.

Answer: The discussion was revised with the reviewers suggestions, including more articles from the literature.

(iii) Page 15: “In this work, the presence of fbsA, fbsB, Srr1 and Srr2 genes were detected in GBS90356 strain (data not shown).” It would be important to include this data as a Supplementary Figure of the manuscript.

Answer: These results are part of a larger study of virulence factor and antimicrobial resistance genes from the GBS90356 strain that have been submitted for publication and are awaiting reviewers comments. I cannot include these data as supplementary.

---

## [Reviewer Report · REVIEWERS COMMENTS]

## REVIEWER #1

Reviewer comments: The authors have satisfied this reviewers comments and the adjustments made in response to all reviewers have strengthened the manuscript.

## REVIEWER #2